# Organ Crosstalk and the Modulation of Insulin Signaling

**DOI:** 10.3390/cells10082082

**Published:** 2021-08-13

**Authors:** Alejandra Romero, Juergen Eckel

**Affiliations:** German Diabetes Center, Institute for Clinical Diabetology, 40225 Düsseldorf, Germany; alejandra.romero@ddz.de

**Keywords:** organ crosstalk, adipokines, myokines, hepatokines, insulin signaling, insulin resistance, extracellular vesicles, exosomes

## Abstract

A highly complex network of organ communication plays a key role in regulating metabolic homeostasis, specifically due to the modulation of the insulin signaling machinery. As a paradigm, the role of adipose tissue in organ crosstalk has been extensively investigated, but tissues such as muscles and the liver are equally important players in this scenario. Perturbation of organ crosstalk is a hallmark of insulin resistance, emphasizing the importance of crosstalk molecules in the modulation of insulin signaling, potentially leading to defects in insulin action. Classically secreted proteins are major crosstalk molecules and are able to affect insulin signaling in both directions. In this review, we aim to focus on some crosstalk mediators with an impact on the early steps of insulin signaling. In addition, we also summarize the current knowledge on the role of extracellular vesicles in relation to insulin signaling, a more recently discovered additional component of organ crosstalk. Finally, an attempt will be made to identify inter-connections between these two pathways of organ crosstalk and the potential impact on the insulin signaling network.

## 1. Introduction

Organ crosstalk represents a system of biological communication that makes it possible for cells in one tissue to send information to cells in another tissue, even at relatively long distances. This process involves molecular, cellular, and neural factors and is a key determinant of physiological homeostasis, with a deep impact on hormonal signaling and action. A complex scenario of crosstalk mediators involving nutrients and metabolites, extracellular vesicles (EVs), and peptides and proteins that may be collectively called “organokines” needs to be considered, and many of these players are known to modify insulin signaling [1,2]. It is presently not clear if these molecules fit into a hierarchy of communication mediators, or if these molecules act independently at different levels of organ crosstalk. Furthermore, multiple organs participate in the crosstalk communication including adipose tissue, skeletal muscle, the liver, the brain, and the gut. On top of this, the intestinal microbiota is also integrated in this system, with some metabolites potentially playing an important role in modifications of insulin signaling [3].

The most detailed understanding of cellular communication and organ crosstalk relates to organokines, which, in this review, we define as classically secreted peptides and proteins. Although secretome analysis data show that, most likely, thousands of peptides and proteins are released by adipocytes, myocytes, liver cells, and other cell types [4,5,6], they can be considered as a relatively homogeneous class of molecules. Many of them are cytokines including interleukins, interferons, chemokines, and similar molecules playing a role in immuno-regulatory processes with either pro-inflammatory or anti-inflammatory functions. The seminal work published by Spiegelman and co-workers nearly 30 years ago identified the cytokine TNFα as a critical signal released by adipose tissue, finally inducing skeletal muscle insulin resistance [7,8]. These findings triggered a tremendous and still ongoing interest in organ crosstalk as a major element of metabolic homeostasis and a key driver for metabolic diseases. In addition to adipokines and myokines, hepatokines have recently gained increasing interest as modulators of insulin signaling [9]. In this review, we aim to focus on organokines that affect the very early steps of insulin signaling, leading to either sensitizing or inhibiting insulin action.

More recent discoveries of novel mediators of organ crosstalk relate to the so-called EVs that include microvesicles and exosomes [10,11]. This novel mechanism of organ crosstalk may also play an important role in the pathogenesis of several diseases including diabetes mellitus [11]. EVs carry a diverse bioactive cargo of proteins, lipids, metabolites, DNA, and RNA (mRNA and small regulatory RNAs). These vesicles are released to the local tissues and the circulation both in a continuous and regulated process [10]. Substantial evidence supports the notion that EVs participate in organ crosstalk and that due to their specific cargo, they may exert profound effects on their target cells. Application of transcriptomic, lipidomic, and proteomic technologies for analysis of EVs from different cell types has resulted in a comprehensive view on the protein, lipid, and nucleic acid content of these vesicles. Specifically, it was shown that adipose tissue constitutes a major source of circulating exosomal miRNAs, leading to the regulation of gene expression in other tissues [12]. Exosomes have also been recognized to be implicated in a spectrum of different liver diseases including hepatocellular carcinoma, NAFLD, and alcoholic liver disease [13]. We here summarize the current knowledge on exosome-mediated modulation of the insulin signaling network including its impact on the pathophysiology of type 2 diabetes mellitus (T2DM). Given that the miRNA cargo may affect the gene expression of organokine targets, it is conceivable that the different pathways of organ crosstalk (secreted proteins vs. exosomes) are inter-connected and may operate in a synergistic way. An attempt will be made to identify such points of convergence, shedding some new light on insulin signaling in the context of organ crosstalk.

## 2. Secreted Modulators of Early Insulin Signaling

Insulin binding to its receptor triggers a highly complex signaling machinery consisting of phosphorylation events and recruitment of insulin receptor substrates, finally resulting in a metabolic and/or mitogenic program (for a recent review, see [14]). Insulin receptor substrate (IRS) 1 and 2 are key elements of this process, being tyrosine phosphorylated by the insulin receptor followed by recruiting downstream signaling effector kinases such as protein kinase b (Akt), S6 kinase, and isoforms of PKC. Modulation of the insulin cascade at the proximal level involving the insulin receptor and IRS1/2 has gained considerable interest for explaining the pathophysiology of lipid-induced insulin resistance [15], pinpointing these mechanisms as key drivers of metabolic dysfunction in obesity and T2DM [16]. Many organokines bind to their own receptors and are able to trigger an intracellular crosstalk with the insulin signaling pathway. As a paradigm, the pro-inflammatory signaling by cytokines such as TNFα and many others has been repeatedly described to negatively modify the insulin signaling pathway at different levels involving serine kinases such as IKKβ and JNK1 [17]. Here, we consider some classically secreted organokines that modulate the insulin cascade at a very early level, meaning the insulin receptor and proximal proteins.

### 2.1. Inhibitory Modulators

Originally described as a natural inhibitor of the insulin receptor tyrosine kinase, fetuin-A (alpha2-HS-glycoprotein) is now considered as a key hepatokine with a substantial impact on the development of insulin resistance in all insulin target tissues [9,18]. Fetuin-A was found to inhibit human insulin receptor autophosphorylation and the tyrosine phosphorylation of IRS1 [19], pointing to an important organ crosstalk molecule with a direct impact on early insulin signaling. In line with that, fetuin-A KO mice exhibit improved insulin signaling, and patients with obesity and NAFLD are characterized by high levels of circulating fetuin-A [9,20,21]. Interestingly, upregulation of fetuin-A can be promoted by elements of metabolic syndrome, namely, increased levels of free fatty acids and glucose, whereas adiponectin is known to reduce the expression of fetuin-A, involving the AMPK pathway [22]. In addition to acting as an early inhibitor of insulin signaling, fetuin-A has also been shown to increase the expression of pro-inflammatory cytokines in adipose tissue and to function as an adaptor protein for saturated fatty acids. This leads to activation of the Toll-like receptor 4, and hence lipid-induced inflammatory signaling and, finally, insulin resistance [18]. Mechanistic studies have shown that fetuin-A does not bind to the α-subunit of the insulin receptor [23]. It was observed that fetuin-A is more likely to bind to tandem fibronectin type 3 (Fn3) domains present within the 194 amino acid residue extracellular moiety of the insulin receptor β-subunit. A mechanism was suggested where insulin binds to the α-subunit of the insulin receptor, leading to a conformational change that promotes the binding of fetuin-A to the β-subunit and receptor inactivation [23]. In addition to fetuin-A, the closely related fetuin-B was also reported to be upregulated in humans with liver steatosis and patients with T2DM [24]. However, data from Stefan and co-workers suggest that only fetuin-A affects insulin signaling despite the significant modulation of glucose homeostasis by these two hepatokines [25].

It may be anticipated that, similar to the liver, other organs could also release proteins capable of modifying insulin receptor activation, finally leading to changes in metabolic performance. A potential candidate is mitsugumin 53 (MG53), a muscle-specific protein with multiple functions including muscle regeneration, calcium homeostasis, excitation–contraction coupling, and myogenesis [26]. In 2013, it was reported that MG53 is upregulated in models of insulin resistance and that it modifies insulin signaling by targeting the degradation of the insulin receptor β-subunit and of IRS1 by its ubiquitin E3 ligase activity [27]. Recently, the same group reported that MG53 is a myokine/cardiokine secreted from striated muscles in response to high glucose or insulin in both rodents and humans [26]. Further, the serum level of MG53 is upregulated in obesity and T2DM, and administration of MG53 was found to inhibit the insulin response in multiple organs. Finally, MG53 was shown to bind to the extracellular domain of the insulin receptor with high affinity, potentially acting as an allosteric inhibitor [28]. These findings define circulating MG53 as a potential target for treating insulin resistance and T2DM. However, the role of MG53 in diabetes has remained highly controversial [26]. Thus, transgenic mice with sustained overexpression of MG53 exhibit an unaltered metabolic function and improved tissue repair capacity [29]. In a very recent study, the authors were not able to reproduce the in vitro effect of MG53 on the insulin response [30]. Importantly, MG53 KO mice were not resistant to high fat-induced obesity and glucose intolerance. However, MG53 plays an important role as a myokine mediating tissue repair and regeneration [31], and future studies will be needed to clarify its impact on systemic metabolic function.

### 2.2. Stimulatory Modulators

The number of organokines that are able to positively modify the early insulin signaling machinery is much more limited compared to crosstalk molecules that lead to perturbation of insulin signaling pathways. This, at least partly, reflects the key role of adipose tissue and the substantial number of pro-inflammatory cytokines released from enlarged adipose tissue mediating insulin resistance. Interestingly, under physiological conditions one of the most abundant adipocyte-secreted factors, namely, adiponectin, exerts a positive crosstalk with the early insulin signaling steps, leading to insulin sensitization [32]. Adiponectin is a multifunctional protein with a broad range of biological functions including regulation of energy metabolism, modulation of inflammatory processes, and improvement in insulin sensitivity. Adiponectin is found in the circulation in multiple forms (low to high molecular weight), and the cellular functions are mediated by two receptors named AdipoR1 and AdipoR2 [33]. The crystal structures of these receptors have been reported as representing a novel class of receptors distinct from G protein-coupled receptors [34]. Downstream signaling of AdipoR involves the adaptor protein APPL1, which is localized in early endosomes and contains several key interactor domains such as the pleckstrin homology and the phosphotyrosine-binding domain [32,35]. In a seminal study, Ryu and co-workers [36] identified the molecular steps leading to a positive modulation of early insulin signaling by adiponectin. It was shown that i) in the basal state, APPL1 forms a complex with IRS1/2, and that ii) insulin or adiponectin led to serine phosphorylation of APPL1 which is then recruited to the insulin receptor, augmenting the downstream signaling pathway. It was further shown that insulin sensitization by adiponectin mediated by APPL1 takes place at the level downstream of the insulin receptor involving IRS1/2. Interestingly, adiponectin appears to function as a downstream effector of the organokine FGF21. This pleiotropic hormone, which is mostly expressed in the liver, has gained considerable interest as an insulin sensitizer and a regulator of carbohydrate and lipid metabolism [12].

In addition to the AdipoRs, adiponectin also binds to T-cadherin, a member of the cadherin family of cell adhesion proteins, being unique due to its C-terminal GPI anchor and lacking a transmembrane intracellular signaling domain [37]. Earlier studies showed that multimers and hexamers of adiponectin accumulate in muscles and the vascular endothelium mediated by T-cadherin [38]. Most importantly, accumulating evidence supports the view that T-cadherin facilitates the accumulation of adiponectin in multivesicular bodies, thereby stimulating the biogenesis and the secretion of exosomes [39]. As pointed out before [10,11], exosomes are now considered as an additional element of intra- and inter-organ crosstalk, with a key role of adipose tissue in this scenario. Interestingly, the systemic level of exosomes decreases substantially upon knockdown of adiponectin or T-cadherin, emphasizing a key regulatory function of adiponectin for this process and highlighting the tight link between classically secreted organokines and the network of exosomal crosstalk. In the subsequent chapter, we consider elements of this crosstalk with regard to modulation of insulin signaling.

## 3. Extracellular Vesicles and Insulin Signaling

Multiple circulating factors are known that modulate inter-organ crosstalk, modifying insulin signaling [40]. Nowadays, the relevance of EVs in this crosstalk has been strongly highlighted. EVs are membrane-enclosed envelopes that carry proteins, lipids, miRNAs, and other factors, which appear to interfere with insulin signaling, promoting insulin resistance [41]. Three different categories of EVs have been described based on their biogenesis: apoptotic bodies, microvesicles, and exosomes, with the last type being of high importance in diabetes mellitus pathology.

Exosomes are generated by invaginations of the plasma membrane, resulting in endocytic vesicles that are released to the extracellular medium [42,43]. Once there, they interact locally or distally with cells from others tissues, inducing changes in cellular function [41,44]. Some of these exosomes seem to regulate insulin sensitivity through two different mechanisms: (i) indirectly by modulating inflammation, or (ii) by direct interaction with insulin-responsive organs. Both mechanisms affect insulin sensitivity through interaction with signaling and downstream molecules, such as phosphatidylinositol-3-kinase (PI3K)/Akt, IRS1, and glucose transporter 4 (GLUT-4), or by mediating the activation of inflammation [44,45]. The defective insulin signaling leads to a lower activity of endothelial nitric oxide synthase (eNOS), with a subsequent lower generation of nitric oxide (NO). Consequently, the reduced activation of this pathway or IRS1/PI3K/Akt signaling results in a lower activity of the mammalian target of rapamycin (mTOR) and additional downstream pathways. This leads to altered cell signaling in response to insulin or to other stimuli in obesity and T2DM [46].

The exosome cargo is tightly involved in the alteration of insulin signaling, making it an important target for studies on the prognosis and diagnosis of metabolic diseases. Most recent studies were focused on the role of miRNAs, short sequences of endogenous RNAs that act as post-transcriptional regulators of gene expression. miRNAs can be secreted freely from different tissues to the extracellular fluids and mediate intercellular communication. However, the miRNAs present in exosomes can be transferred to cells, avoiding their degradation and interfering with their intracellular messenger RNA (mRNAs) mediating the inter-organ crosstalk [47]. Additionally, some of them are capable of binding to various proteins, including lipoproteins, or argonaute proteins, in order to form silencing complexes and alter the intracellular pathway of insulin signaling [48]. Consequently, some of these miRNAs have been described to regulate the expression of hepatic genes such as IRS1 and the peroxisome proliferator-activated receptor gamma (PPAR-γ) or interfere with important targets such as AKT-interacting protein (AKTIP) in skeletal muscle cells, among others [45,48].

Several exosomal miRNAs are dysregulated in patients with diabetes, affecting different metabolic pathways. On one hand, some exosomal miRNAs negatively alter insulin signaling. For example, miRNA-362, miRNA-30, or miRNA 150-5p, present in T2DM exosomes, interact with the mTOR substrate, reducing its activity and altering the cell signaling in response to insulin; however, their origin is unknown [49,50]. On the other hand, some exosomal miRNAs can interact with insulin signaling, and they do not just have deleterious effects. Xu et al. described that miR-26a from pancreatic exosomes is capable of increasing the insulin sensitivity in the liver, but the levels of this miRNA are diminished in obese humans and mice, being a possible target for the treatment of this disease [51]. Therefore, the study of the exosome cargo is an important strategy to prevent and treat several metabolic diseases. In this chapter, we focus on exosomal miRNAs from principal tissues such as adipose tissue, muscles, the pancreas, and the liver in the context of potential cell targets and the modulation of insulin signaling. To sum up, the main miRNAs are included in Figure 1, providing an overview of selected exosomal miRNAs and their effects on different targets tissues in the context of insulin signaling (see also Table 1). All miRNAs shown in Figure 1 have been confirmed to be transferred by extracellular vesicles.

### 3.1. Adipose Tissue Exosomal miRNAs

Adipose tissue constitutes an important source of exosomes and, in particular, exosomal miRNAs that can regulate gene expression in distant tissues such as the liver or skeletal muscle [40,47]. These exosomal miRNAs are being extensively studied for their involvement in insulin resistance and in the regulation of systemic metabolism in obese patients by connecting different organs [52,53].

One of the principal targets of adipose tissue exosomes is the liver. Several studies have described the release of miR-99b or miR-141-3p from fat depots taken up by this organ regulating insulin pathways. miR-99b is able to regulate FGF-21 expression by binding to the three prime untranslated region (3′UTR) of its mRNA in the liver. This growth factor is known to improve insulin sensitivity and reduce lipid accumulation through repression of rapamycin complex 1 (mTORC1) in hepatocytes, influencing whole-body metabolism. Importantly, in obese patients, the levels of this miRNA are elevated and promote insulin resistance and adipogenesis by repression of FGF-21 [12,54,55]. Nevertheless, not all miRNAs from adipose tissue have deleterious effects on insulin signaling in the liver. miR-141-3p is responsible for normal Akt phosphorylation upon insulin stimulation. However, its levels are reduced in obese adipocyte exosomes and lead to impaired insulin signaling and glucose uptake in hepatocytes [44,56].

Some miRNAs from adipose tissue also regulate lipid metabolism and insulin signaling in skeletal muscle cells. The skeletal muscle represents about 40% of the body mass, being responsible for almost 80% of insulin-stimulated glucose uptake in healthy subjects, and is considered the major site of peripheral insulin resistance [57]. Obese mice show high levels of exosomal miR-130, which is correlated with BMI. This miRNA regulates lipid catabolism in skeletal muscle to inhibit the expression of the peroxisome proliferator-activated receptor gamma coactivator 1-alpha (PGC1α) that promotes mitochondrial fatty acid oxidation and GLUT-4 expression and increases insulin-stimulated glucose transport [44,54,58]. Other miRNAs described in these exosomes are miR-227a and miR-27a. These molecules are capable of decreasing the expression of GLUT-4 by downregulation of PPAR-γ signaling and alteration of PI3K/Akt activation in skeletal muscle cells and adipocytes, leading to insulin resistance [52,59,60]. In addition, miR-27a has been described to mediate macrophage infiltration and activation of nuclear factor kappa B (NF-κB), inducing inflammation in skeletal muscle cells, finally affecting insulin signaling by alteration of IRS1 and GLUT-4 expression [56,60,61].

In addition to regulating GLUT-4 or insulin receptor expression, other pathways in skeletal muscle cells are affected by miRNAs from adipose tissue exosomes. Kukreti and Amuthavalli showed in aging mice that miR-34a acts as a regulator of insulin signaling by targeting the ceramide kinase (CERK) [62]. The CERK induction antagonized insulin signaling to block the activation of the anabolic enzyme Akt/PKB and also activate the JNK pathway that interferes with IRS1, mitigating the expression of GLUT-4 [60,63]. Therefore, exosomal miRNAs such as miR-34a or miR-27a may be considered as key mediators of obesity-induced inflammation and insulin resistance.

An important resource of exosomal miRNAs in obesity is macrophages that have infiltrated adipose tissue. Adipose tissue macrophages (ATMs) contribute to the development of insulin resistance by the release of several pro-inflammatory mediators, which can block insulin action in adipocytes and other tissues. In addition to these molecules, ATMs mediate organ crosstalk through the release of exosomal miRNAs, modulating insulin signaling in distal tissues such as the liver or skeletal muscle. One of the main exosomal miRNAs released by ATMs is miR-155, which exhibits effects in several tissues [11]. In the liver, miR-155 can inhibit insulin signaling and affect the insulin/glucose tolerance through suppression of its target gene PPAR-γ. This effect has also been described in skeletal muscle cells, adipocytes, and the pancreas. Although PPAR-γ expression is not high in the liver, its inhibition affects the insulin signaling cascade by increasing gluconeogenesis [64]. Nevertheless, this gene is highly expressed in adipocytes, where PPAR-γ plays an important role in glucose homeostasis and adipocyte differentiation. In addition, in adipose tissue, miR-155 promotes insulin resistance by downregulation of GLUT-4, decreasing the tyrosine phosphorylation of the insulin receptor and of IRS1, as well as attenuating the activation of Akt/PKB [64,65,66,67].

In muscle tissues, some effects are similar, where miR-155 mitigates the IRS1-associated PI3K activity and Akt and decreases Akt phosphorylation [11,64]. Some publications described that miR-155 in the pancreas impaired insulin secretion and increased β-cell proliferation by v-maf musculoaponeurotic fibrosarcoma oncogene family, protein B (MAFB) repression [53,68]. Overall, exosomal miRNAs from adipose tissues and ATMs exert inter- and intra-organ crosstalk and modulate insulin signaling at different levels including the gene expression of critical downstream effectors such as GLUT-4.

### 3.2. Muscle Exosomal miRNAs

The exosomal miRNAs derived from skeletal muscle have been called myomiRs. There are several myomiRs differently expressed in patients with diabetes and non-diabetic patients [69]. For example, it has been described that miR-133a is inversely correlated with the expression of uncoupling-protein-2 (UCP2) in pancreatic islets. This protein is involved in the glucose-stimulated insulin secretion (GSIS) by the control of reactive oxygen species (ROS) production, and its lower expression in T2DM patients promotes insulin resistance [69,70]. Other myomiRs, highly expressed in these patients, have also been described to alter pancreatic insulin release by different mechanisms. On the one hand, myomiR-7 reduces insulin release by mitigation of β-cell proliferation, and myomiR-96 does so by attenuation of the RabGTPase effector granuphilin, a component of secretory granules in β-cells [71,72]. Likewise, Baroukh and collaborators described that myomiR-124a was also capable of attenuating insulin release to affect the transcription factor involved in β-cell differentiation, PDX1 [73]. All of these miRNAs and others exert negative effects on the insulin pathway, affecting insulin secretion or signaling in other organs [69,74]. On the other hand, myomiRs such as myomiR-26, myomiR-148, myomiR-182, and myomiR-483 are capable of stimulating insulin gene transcription and insulin secretion; however, their levels are low in the exosomes of diabetic patients. Therefore, the release of these exosomes could be a potential target in the treatment of these patients [69,74,75].

The effects of some myomiRs have also been described in other tissues. Some of them can exert an autocrine effect and alter insulin signaling. Katayama and colleagues [48] described that miR-20b-5p promoted insulin resistance in skeletal muscle by inhibiting Akt expression and the activation of the signal transducer and activator of transcription 3 (STAT3). STAT3 induces the expression of Fbxo40, a muscle-specific E3 ubiquitin ligase that stimulates ubiquitin conjugation, leading to degradation of IRS1 [11,48,76]. Some authors such as Castaño et al. or Barlow et al. have described that exercise could change the expression of these miRNAs, improving insulin sensitivity [69,77]. Consequently, the knowledge of exosomes and their targets could serve as a potential therapy in metabolic syndrome.

### 3.3. Pancreatic Exosomal miRNAs

The pancreas is the fundamental organ for the body’s blood glucose control. Pancreatic β-cells are responsible for secreting insulin in response to glucose. However, they are also capable of releasing factors that modulate glucose homeostasis by controlling insulin secretion and sensitivity and the insulin signaling pathways in several tissues [51,78]. As with the liver or muscles, β-cells are capable of modulating insulin pathways in different tissues by exosomal miRNAs.

Some studies have described alterations in the insulin sensitivity of muscle cells caused by exosomal miRNAs from β-cells. The main mechanism affected in the muscle cells is the expression of GLUT-4. miRNAs such as miR-450b-3, miR-883b-5p, miR-666-3p, or miR-151-3p were demonstrated to inhibit insulin signaling in myotubes. These miRNAs abolish PI3K/Akt signaling and GLUT-4 trafficking but preserve Forkhead box protein O1 (FoxO1) induction [11,59]. On the other hand, miRNAs such as miR-26s were shown to decrease insulin sensitivity by impairing actin cytoskeleton remodeling and regulating gene expression [51,56]. In addition to regulating the expression of GLUT-4 or the insulin signaling of muscle cells, some exosomal miRNAs from β-cells have demonstrated effects in other tissues. For example, miRNA-223 stimulates the expression of GLUT-4 in muscle cells and cardiomyocytes. Interestingly, their levels are decreased in metabolic syndrome, being an important target in the development of diabetes mellitus [56].

Exosomal miRNAs derived from β-cells do not exclusively exert effects on muscle cells. Xu and collaborators demonstrated that miR-26a was capable of preventing obesity-induced metabolic abnormalities in the liver, and insulin resistance locally and distally. However, miR-26 levels are significantly reduced in obese humans and mice [51]. Moreover, in the liver, miR29s was shown to regulate hepatic insulin sensitivity and to control glucose homeostasis. The target of this miRNA is the inhibition of the p85α subunit of PI3K. Consequently, miR-29s interacts with IRS1 and PI3K/Akt and contributes to insulin resistance [79,80].

### 3.4. Liver miRNA Exosomes

The liver is a dynamic organ that plays critical roles in many physiological processes, including the regulation of systemic glucose and lipid metabolism. As with the pancreas or adipose tissue, it is a secretory organ capable of releasing miRNA encapsulated in exosomes into the blood, mediating organ crosstalk. Some of these miRNAs exert effects directly in the adjacent cells, promoting insulin resistance in the liver. This is the case for miR-128-3p, miR-122, and miR-192 that have been described to abolish the PPAR-γ signaling [40,56]. Furthermore, miRNAs from liver exosomes can interact with other tissues mediating organ crosstalk. The hepatic exosomal miR-130a-3p has recently been described to participate in lipid and glucose metabolism through adipocyte interaction. This miRNA downregulates the expression of fatty acid synthetase (FASN) and PPAR-γ in the adipocytes, suppressing adipogenesis. In addition, miR-130a-3p interacts with the PH domain leucine-rich repeat protein phosphatase 2 (PHLPP2), increasing the levels of p-AKT and p-AS160, and finally promoting GLUT4 translocation [56,81]. Currently, some exosomal miRNAs from the liver have been described that affect insulin signaling; however, future studies are required to clarify their potential impact as a target for therapeutic strategies.

### 3.5. Exosomal miRNA and Classical Modulators in Metabolic Diseases

In the context of metabolic diseases, crosstalk studies have been focused on the role of organokines, lipids, and other mediators released from adipose tissue, the liver, or the pancreas. These modulators exert well-known effects on metabolism. Nonetheless, in recent years, exosomal miRNAs have received increasing attention due to their complex gene regulatory functions in the context of organ crosstalk. Table 1 summarizes the functional impact of exosomal miRNAs on early and downstream insulin action.

In this review, we have described the interaction of exosomal miRNAs directly with insulin signaling. However, they can also alter the level of organokines in the organ crosstalk scenario. Santovito et al. demonstrated that miR-326, encapsulated in macrophage exosomes, is inversely correlated with its putative target adiponectin [82]. They demonstrated that the levels of miR-326 in T2DM patients were increased, but the expression of AdipoR2 and adiponectin was reduced. Therefore, the high levels of circulating miR-326 represented a novel modulator of the adiponectin pathway [82,83]. Similar to that, Thomou et al. reported that miR-99b affected the expression of FGF-21, a growth factor that acts as a protective adipokine, hepatokine, and myokine, with paracrine and autocrine effects in energy metabolism [12,60]. Similarly, miR-27a coming from ATMs affects these classical modulators by promoting the expression of NF-κB in adipose tissue, and the release of inflammatory cytokines that alter insulin signaling indirectly [60]. Nevertheless, other effects than those of exosomal miRNAs on these modulators have been described. A recent study showed that adiponectin stimulates exosome biogenesis through T-cadherin binding and decreases its content of ceramides, which promote insulin resistance [37]. These considerations show that classically secreted modulators and the exosomal machinery are linked and functionally both relevant to orchestrate the organ crosstalk machinery. Future studies should use a more integrated approach to improve our understanding of how these different layers of organ communication cooperate, and how the insulin signaling network is controlled by these pathways. This may open a new perspective on states of insulin resistance as a result of dysregulated organ crosstalk.

## 4. Conclusions

The insulin signaling network comprises a complex machinery consisting of a wide range of genes and proteins that regulate metabolic homeostasis. In this review, we summarized novel molecules derived from organ crosstalk that modify the insulin signaling network at different levels. Classical protein modulators, such as adiponectin or the recently described fetuin-A, influence the early steps of insulin signaling by affecting the insulin receptor or IRS1. In addition, the role of exosomal miRNAs has gained considerable interest, revealing novel molecules that modify downstream insulin signaling. These crosstalk mediators are becoming increasingly important regarding their role as modulators and as blood biomarkers in disease progression. In addition, recent studies have described the potential interaction between these classical modulators and exosomal miRNAs in the context of metabolic diseases. Thus, exosomal miRNAs not only have a direct impact on insulin signaling but could also have an effect on the organ crosstalk machinery. Therefore, increasing our knowledge on exosomal miRNA targets and their role in organ crosstalk will be instrumental in gaining an advanced understanding of metabolic diseases, and in the development of new therapeutic strategies.

## Figures and Tables

**Figure 1 cells-10-02082-f001:**
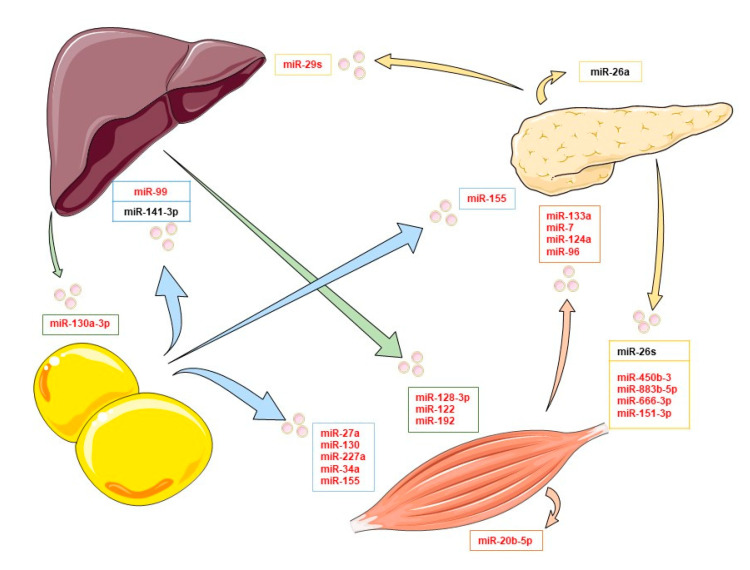
Exosomal miRNAs and their organ target in the insulin signaling context. In this image, some of the main exosomal miRNAs implicated in organ crosstalk affecting insulin signaling are collected. Arrows indicate the exosomal miRNAs released by adipose tissue (blue), skeletal muscle cells (orange), the pancreas (yellow), and the liver (green). Inhibitory exosomal miRNAs are shown in red, and inductors are shown in black.

**Table 1 cells-10-02082-t001:** Effect of exosomal miRNAs on early and downstream insulin actions.

Functional Impact	Target	Target Organ	Exosomal miRNA	Effect
Insulin receptor and PI3K signaling pathway	IRS1 and PI3K/p-Akt	Liver, Muscle	miR 29s, miR 450b-3, miR-227a, miR-27a, miR 883b-5p, miR 666-3p, miR 151-3p, miR-155	↓
	MAFB	Pancreas	miR-155	↓
	Akt phosphorylation	Liver	miR- 141-3p	↑
	AKTIP/STAT3	Muscle	miR-20b-5p	↓
GLUT-4 translocation and glucose homeostasis	PI3K/p-Akt and GLUT-4 trafficking	Muscle	miR-450b-3, miR-883b-5p, miR-666-3p, miR-151-3p	↓
	CERK	Muscle	miR-34a	↓
	IRS phosphorylation	Muscle	miR-155	↓
	PGC1α	Muscle	miR-130	↓
	PPAR-γ	Adipose tissue, Muscle, Liver	miR-130a-3p, miR 128-3p, miR-155, miR 122, miR 192, miR-227a, miR-27a	↓
Organokine expression	FGF-21	Liver	miR-99b	↓
	Adiponectin	Adipose tissue	miR-326	↓

## Data Availability

Not applicable.

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
