# Peer review of "Organ Crosstalk and the Modulation of Insulin Signaling"

_cells, 2021, doi:10.3390/cells10082082_

Round 1
Reviewer 1 Report
The authors summarize recent work on organ crosstalk in insulin signalling with a special focus on the role of extracellular vesicle microRNA. The topic is of interest and the manuscript is pleasant to read. Some comments about the review are detailed below:
- The authors should take care to properly cite original research studies and avoid repeated listing of earlier reviews on the subject.
- Circulating microRNA does not necessarily mean extracellular vesicles enclosed microRNA, and if so, the origin of the vesicles is not always precisely determined. For instance, on page 6, line 267, the authors state that ATM are the “main resource of exosomal miRNAs in obesity”. This assumption is certainly an overstatement given the relative abundance of ATM.
- Page 5 line 245, the authors discuss quantitative aspects of skeletal muscle insulin-uptake and skeletal muscle contribution to insulin resistance. It would be interesting to extent this kind of quantitative considerations to adipose tissue/endocrine pancreatic tissues to discuss the physiological relevance of exosomal miRNA-released by these tissues.
- Page 8, line 375: “So, the high levels of circulating miR-326 represented a novel modulator of the adiponectin pathway”. The causal link remains unclear, in particular in light of the lack of specificity of circulating microRNA as biomarkers (Witwer, Clinical Chemistry, 2015).
- Figure 1: the authors should indicate for which microRNAs transfer by extracellular vesicles has clearly been demonstrated
Minor corrections:
- line 37: grammatical structure “organokines, which describes…” should be revised
- Iine 69 “feasible” should probably be replaced by “conceivable”
Author Response
We thank this reviewer for the positive evaluation of our manuscript and the critical and helpful suggestions.
- The authors should take care to properly cite original research studies and avoid repeated listing of earlier reviews on the subject.
In this revised version, we have included new original studies and studies related to the topic.
- Circulating microRNA does not necessarily mean extracellular vesicles enclosed microRNA, and if so, the origin of the vesicles is not always precisely determined. For instance, on page 6, line 267, the authors state that ATM are the “main resource of exosomal miRNAs in obesity”. This assumption is certainly an overstatement given the relative abundance of ATM.
We agree with this reviewer and have changed the wording appropriately.
- Page 5 line 245, the authors discuss quantitative aspects of skeletal muscle insulin-uptake and skeletal muscle contribution to insulin resistance. It would be interesting to extent this kind of quantitative considerations to adipose tissue/endocrine pancreatic tissues to discuss the physiological relevance of exosomal miRNA-released by these tissues.
This reviewer states that we discuss quantitative aspects of insulin uptake by skeletal muscle. This is incorrect since our discussion on p. 5 (line 245) refers to glucose uptake. Given the key role of skeletal muscle for insulin-stimulated glucose uptake, we prefer not to extend these considerations to other tissues.
- Page 8, line 375: “So, the high levels of circulating miR-326 represented a novel modulator of the adiponectin pathway”. The causal link remains unclear, in particular in light of the lack of specificity of circulating microRNA as biomarkers (Witwer, Clinical Chemistry, 2015).
In this review, we discuss several miRNAs with potential effects on the insulin signaling pathway. Some of them are also useful as biomarkers in several diseases such as diabetes mellitus. Recently, Sabry et al. have described the correlation between miR-326 and adiponectin levels in these patients. Type I and II diabetes patients have higher levels of this miRNA than control subjects. However, their adiponectin levels are lower. Santovito et al. also described this in 2014. They demonstrated that miR-326 interacts with adiponectin receptor (ADIPOR)-1, ADIPOR-2, and APPL-1, involved in the intracellular pathway of insulin signaling, and affects the adiponectin levels. Therefore, exosomal miRNA studies could be useful as potential biomarkers, and for understanding the interaction with classical modulators like in this case.
Sabry, et al. 2020. European Journal of Molecular & Clinical Medicine; 7 (6).
Santovito, D et al 2014. J Clin Endocrinol Metab. 99, E1681-E1685.
- Figure 1: the authors should indicate for which microRNAs transfer by extracellular vesicles has clearly been demonstrated
All miRNAs shown in Fig. 1 have been confirmed to be transferred by extracellular vesicles. This is now mentioned in the MS on p. 5, line 225.
Minor corrections
Done
Reviewer 2 Report
In their review manuscript entitled “Organ Crosstalk and the Modulation of Insulin Signaling”, Romero and Eckel summarize organokines whose effects target the insulin receptor or IRS1/2, as well as miRNA’s released by extracellular vesicles. Generally, their review was thoughtful, well-sourced, and is suitable for publication in “Cells”. Below, I have listed a few minor comments:
- In their discussion of organokines, the authors focus on “early steps” of insulin signaling-however this term is ambiguous (early temporally in the signaling cascade? or effecting proteins proximal to the insulin receptor?). While they are not explicit in their definition, I believe they mean steps that affect the insulin receptor and IRS1/2. Making this definition earlier and more explicit would prevent confusion on the part of the reader.
- In their selection of organokines to discuss, the authors focused on Fetuin-A, the controversial MG53, and adiponectin. While they do not claim to be comprehensive, the authors may also want to at least mention FGF21, which by activating FAO decreases lipid deposition and thereby may reduce PKC-related inhibition of the insulin receptor and/or IRS1 and is mentioned at the end of the manuscript, or RBP4 which does not directly affect the insulin receptor or IRS1/2 but does affect ATM’s which are discussed later in the manuscript.
- There are a few times when the authors use the term “organokines” ambiguously-it usually refers specifically to peptides and proteins, but occasionally seems to refer to lipids and other secreted molecules as well (ex. Line 367, where the term “organokines” refers to the previous sentence that also mentions lipids with proteins and peptides).
- The connection between organokines and EV-released miRNA’s is very intriguing. It would be helpful if this connection was fleshed out, perhaps by integrating into figure 1, or as a separate figure.
Author Response
We thank this reviewer for the positive evaluation of our manuscript and the critical and helpful suggestions.
- In their discussion of organokines, the authors focus on “early steps” of insulin signaling-however this term is ambiguous (early temporally in the signaling cascade? or effecting proteins proximal to the insulin receptor?). While they are not explicit in their definition, I believe they mean steps that affect the insulin receptor and IRS1/2. Making this definition earlier and more explicit would prevent confusion on the part of the reader.
We thank this reviewer for the comment. A clear definition of the term "early steps" of insulin signaling has been included on p. 2, line 160.
- In their selection of organokines to discuss, the authors focused on Fetuin-A, the controversial MG53, and adiponectin. While they do not claim to be comprehensive, the authors may also want to at least mention FGF21, which by activating FAO decreases lipid deposition and thereby may reduce PKC-related inhibition of the insulin receptor and/or IRS1 and is mentioned at the end of the manuscript, or RBP4 which does not directly affect the insulin receptor or IRS1/2 but does affect ATM’s which are discussed later in the manuscript.
We appreciate this suggestion and mention FGF21 on p. 4, line 160.
- There are a few times when the authors use the term “organokines” ambiguously-it usually refers specifically to peptides and proteins, but occasionally seems to refer to lipids and other secreted molecules as well (ex. Line 367, where the term “organokines” refers to the previous sentence that also mentions lipids with proteins and peptides).
Thank you for the comment, we have checked the text and corrected this mistake. Certainly, when we talk about” organokines”, we refer to peptides and proteins.
- The connection between organokines and EV-released miRNA’s is very intriguing. It would be helpful if this connection was fleshed out, perhaps by integrating into figure 1, or as a separate figure.
Thanks for your recommendation. We have split Figure 1 in a Figure and a table for clarifying these relationships. The effect of miRNA on organokine expression is highlighted in the Table.